# Loop-Mediated Isothermal Amplification (LAMP) as a Promising Point-of-Care Diagnostic Strategy in Avian Virus Research

**DOI:** 10.3390/ani12010076

**Published:** 2021-12-30

**Authors:** Faiz Padzil, Abdul Razak Mariatulqabtiah, Wen Siang Tan, Kok Lian Ho, Nurulfiza Mat Isa, Han Yih Lau, Jalila Abu, Kuo-Pin Chuang

**Affiliations:** 1Laboratory of Vaccine and Biomolecules, Institute of Bioscience, Universiti Putra Malaysia, Serdang 43400, Selangor, Malaysia; faizpadzil@gmail.com (F.P.); wstan@upm.edu.my (W.S.T.); nurulfiza@upm.edu.my (N.M.I.); 2Institute for Medical Research, National Institute of Health, Setia Alam, Shah Alam 40170, Selangor, Malaysia; 3Department of Cell and Molecular Biology, Faculty of Biotechnology and Biomolecular Sciences, Universiti Putra Malaysia, Serdang 43400, Selangor, Malaysia; 4Department of Microbiology, Faculty of Biotechnology and Biomolecular Sciences, Universiti Putra Malaysia, Serdang 43400, Selangor, Malaysia; 5Department of Pathology, Faculty of Medicine and Health Sciences, Universiti Putra Malaysia, Serdang 43400, Selangor, Malaysia; klho@upm.edu.my; 6Biotechnology and Nanotechnology Research Centre, Malaysian Agricultural Research and Development Institute (MARDI), Persiaran MARDI-UPM, Serdang 43400, Selangor, Malaysia; hylau@mardi.gov.my; 7Department of Veterinary Clinical Studies, Faculty of Veterinary Medicine, Universiti Putra Malaysia, Serdang 43400, Selangor, Malaysia; jalila@upm.edu.my; 8International Degree Program in Animal Vaccine Technology, International College, National Pingtung University of Science and Technology, Pingtung 912, Taiwan; kpchuang@g4e.npust.edu.tw; 9Graduate Institute of Animal Vaccine Technology, College of Veterinary Medicine, National Pingtung University of Science and Technology, Pingtung 912, Taiwan; 10School of Medicine, College of Medicine, Kaohsiung Medical University, Kaohsiung 807, Taiwan; 11Research Center for Animal Biologics, National Pingtung University of Science and Technology, Pingtung 912, Taiwan; 12School of Dentistry, Kaohsiung Medical University, Kaohsiung 807, Taiwan

**Keywords:** avian, virus, diagnostic, LAMP, rapid

## Abstract

**Simple Summary:**

Many of the existing screening methods of avian viruses depend on clinical symptoms and pathological gross examinations that still necessitate confirmatory microscopic testing. Confirmation of a virus is often conducted at centralized laboratories that are well-equipped with instruments for virus isolation, hemagglutinin inhibition, virus neutralization, ELISA, PCR and qPCR. These assays are known for their great accuracy and sensitivity, and hence are set as standard practices. Nevertheless, limitations arise due to the time, cost and on-site applicability. As the technology progresses, molecular diagnostics should be more accessible to isolated areas and even practicable for use by non-skilled personnel such as farmers and private breeders. One of the point-of-care diagnostic strategies to consider for such matters is loop-mediated isothermal amplification (LAMP).

**Abstract:**

Over the years, development of molecular diagnostics has evolved significantly in the detection of pathogens within humans and their surroundings. Researchers have discovered new species and strains of viruses, while mitigating the viral infections that occur, owing to the accessibility of nucleic acid screening methods such as polymerase chain reaction (PCR), quantitative (real-time) polymerase chain reaction (qPCR) and reverse-transcription qPCR (RT-qPCR). While such molecular detection methods are widely utilized as the benchmark, the invention of isothermal amplifications has also emerged as a reliable tool to improvise on-field diagnosis without dependence on thermocyclers. Among the established isothermal amplification technologies are loop-mediated isothermal amplification (LAMP), recombinant polymerase amplification (RPA), strand displacement activity (SDA), nucleic acid sequence-based amplification (NASBA), helicase-dependent amplification (HDA) and rolling circle amplification (RCA). This review highlights the past research on and future prospects of LAMP, its principles and applications as a promising point-of-care diagnostic method against avian viruses.

## 1. Introduction

The loop-mediated isothermal amplification method (LAMP) is distinguished by the utilization of at least four different primers which specifically recognize six distinct regions on the target nucleotide sequence [1]. The LAMP reaction is facilitated by unique *Bst* polymerase with strand displacement activity which does not require denaturation of the double-stranded DNA during the amplification process. Since the heating of nucleic during the denaturation step is inessential, reaction can proceed optimally at a constant temperature of around 60 °C to 70 °C in a single step reaction [2]. While the alignment of the LAMP primers at six target regions in one reaction further increases the specificity of the assay, the reaction of this method is presented to be fast and much simpler without reliance on expensive equipment. Using the LAMP method, positive results can be recorded in as little as 20 to 30 min in a single-step reaction [1,3]. The detection of RNA can also be achieved by adding reverse transcriptase to the reaction, or using isothermal polymerase with reverse transcription activity for reverse-transcription LAMP (RT-LAMP).

Provided with such advantages, the LAMP method has been widely applied for the screening of many diseases and pathogens including avian viruses. Most of the pathogen screenings are directed toward poultry, which provides a major food source for mankind. Coordinated disease management and control in this industry are very crucial in order to maintain a continuous and healthy supply of meat, eggs and poultry products. In worst-case scenarios, disease outbreaks of highly pathogenic viruses such as avian influenza adversely affect economies and food security throughout the pandemic period due to severe clinical symptoms and high mortality rates. The spread of avian pathogenic viruses has also caused financial loss to the industry of domesticated birds, from the selective breed of racing pigeons to exotic species of parrots. In notable incidents, humans have contracted several subtypes of avian influenza viruses sporadically following the infection of birds, causing severe illness and even death [4,5,6].

The rapid spread of highly pathogenic avian viruses has raised major concerns due to the need for a quick detection system. The LAMP method is proposed as a convenient solution via point-of-care testing, i.e., an analysis that is conducted on-site or close to the desired setting or patient. Point-of-care testing was previously dominated by biosensor and lateral flow strips or dipsticks using immobilized labels and antibodies. Nonetheless, emerging utilization of polymerase chain reaction (PCR) testing has led to the development of several isothermal amplification technologies such as LAMP, recombinant polymerase amplification (RPA), strand displacement activity (SDA), nucleic acid sequences-based amplification (NASBA), helicase-dependent amplification (HDA) and rolling circle amplification (RCA). Nucleic acid-based systems are considered simpler, cost-efficient, fast, accurate, reliable and easy to operate even by non-technical personnel [7]. These are anticipated merits of an ideal point-of-care diagnostic strategy. Through the establishment of these methods, many common avian viruses have been successfully screened via clinical studies with good limits of detection and efficient sensitivity.

## 2. Origin and Advancements of the LAMP Method

The discovery of the LAMP method by Notomi et al. [1] dates back more than two decades, and thereafter was patented by Eiken Chemical Co., Ltd. in Tokyo, Japan [3]. The gene amplification method was developed primarily based on the application of the uniquely designed primers that function together in a set. The main LAMP primers consist of a forward outer primer and backward outer primer called F3 and B3, respectively, and the key binding primers that form the stem loops, creating a structure called the forward inner primer (FIP) and backward inner primer (BIP), as depicted in Figure 1. The optional addition of loop primer forward (LF) and loop primer backward (LB) serve as starting materials to enhance the DNA synthesis, and thus reduce the amplification time [8]. The duration of the LAMP reaction can be further shortened by the inclusion of a forward swarm primer and backward swarm primer that target the upstream region of the FIP and BIP binding sequences [9].

Prior to the LAMP primer design, any target sequence should be analyzed to determine the region to which the primer set would align; either to a specific strain or a conserved sequence within closely related species. Multiple sequence alignment (MSA) of available genomes or published sequences identifies conserved regions, which serve as the template for the primer design. A concise LAMP primer design is crucial to generate a set of functional primers that would perform at optimum level under the feasible conditions. The features that should be taken into measure are oligonucleotide length, compositions of guanine and cytosine (G-C content), melting temperature and the stability of the primer ends. For the LAMP primers, F3 and B3 should have a length of 18 to 20 nucleotides, while both FIP and BIP are set to have a functional length between 38 and 42 nucleotides [1]. The G-C content LAMP primer set would recognize a target region of within 300 nucleotides, short enough to detect small conserved regions in many genes. The GC content in LAMP oligonucleotides, like most primers, is selected to be between 50% and 60% for GC-normal and GC-rich regions. For the melting temperature (Tm) of the oligonucleotides, the value can be predicted using the nearest neighbour method to obtain a value around 55 °C to 60 °C for an AT-rich region and 60 °C to 65 °C GC-rich region. Another key feature in LAMP primer design is the stability of the oligonucleotide ends. The LAMP primer stability is calculated via the delta G value, in coherence with the standard PCR primer design to be less than −4 kcal/mol [10].

While each parameter in developing the primers needs to be addressed precisely, the design steps have been simplified using the available online software. The first and most widely used online software dedicated to LAMP primer design is Primer Explorer (https://primerexplorer.jp/e/ accessed on 1 September 2021), developed by Eiken Chemical Co., Ltd. (Tokyo, Japan). The free-access software can generate multiple sets of LAMP primers based on the input target sequence with automated yet adjustable primer designing parameters. Besides, several LAMP-designing primer software packages have also been created to ease research in developing the LAMP method for a wide range of applications; (e.g., NEB, https://lamp.neb.com/#!/ accessed on 1 September 2021; Optigene, http://www.optigene.co.uk/products-primer-design-service/ accessed on 1 September 2021; and PREMIER Biosoft, http://www.premierbiosoft.com/isothermal/lamp.html accessed on 1 September 2021).

### 2.1. Strand Displacement Bst DNA Polymerase: The Key Element

The key factor in the development of LAMP is the unique function of *Bst* DNA polymerase, an enzyme with high strand-displacement activity that facilitates the amplification in isothermal conditions. From the perspective of conventional PCR, the denaturation step is required to break the hydrogen bonds of the double-stranded DNA at high temperature (90 °C to 99 °C). Once the template DNA has been denatured into single strands, the DNA polymerase binds to the single-stranded DNA and proceeds with addition of free nucleotides during the extension step [1,11]. The *Bst* DNA polymerase, isolated from *Geobacillus stearothermophilus*, has a helicase-like activity that unwinds the double-stranded template DNA by itself and would not be inhibited by secondary structure of DNA at a constant temperature of around 60 °C to 70 °C. This enzyme is able to serve as a platform to develop an isothermal screening system, especially in the diagnostic field, in which a molecular detection can be performed at minimal cost in various settings with efficient portability. One unique feature of the LAMP method is the production of concatemers, which are DNA molecules of the same sequence in repeats and chained together continuously [12]. Concatemers are usually the result of RCA [13]. These concatemers of the amplified target gene following the LAMP reaction are expected to be visible on agarose gel electrophoresis as ladder-like bands for all positive results. The yield of the LAMP reaction is also affected by betaine as an additional component in the reaction mixture at varying degrees. In several LAMP developments, betaine is used to enhance the amplification rate by destabilizing the DNA helix and dissolving the formation of the secondary structure, especially in GC-rich regions [1,14].

This thermostable polymerase enzyme, however, does not possess the 5′ to 3′ exonuclease activity to proofread the amplified DNA, while RNA cannot be directly amplified as a target for LAMP [11,15]. The reverse transcription of the RNA can be achieved by simply adding the reverse transcriptase into the LAMP reagents to produce the complementary DNA (cDNA) as a target precursor in a single-step reaction of the reverse transcriptase LAMP (RT-LAMP) method [8,16]. In a recent breakthrough, *Bst* 3.0 DNA polymerase (NEB, Hitchin, UK) and OmniAmp polymerase (Lucigen, Middleton, WI, USA) have been designed in silico with efficient reverse transcription activity which readily amplifies RNA samples during a LAMP reaction [17,18]. Meanwhile, the inclusion of an enzyme with proofreading ability in a LAMP reaction is aimed to overcome the potential mismatch of primers against the target sequence with high genetic diversity, usually from the viral genomes of multiple subtypes and variants [19]. The use of a Q5 high-fidelity DNA polymerase (NEB, Hitchin, UK) increases the sensitivity of the LAMP mismatch-tolerant method [19,20,21].

### 2.2. Optimizations in LAMP Method

Two major parameters should be accounted for to assess the feasibility of the LAMP reaction in detecting the gene of interest, which are incubation temperature and reaction time. The optimality of the incubation temperate is dependent on the hybridization of the four main LAMP primers used (F3, B3, FIP and BIP) to target the nucleotides in the target sequence. Since the length and composition of the nucleotide bases of each primer would affect the melting temperature, the LAMP primers, especially the F2 and B2 sequences in FIP and BIP, respectively, are designed to have a melting temperature within the same range as the recommended temperature for the *Bst* polymerase to catalyze the reaction effectively [1]. In this case, the same set of LAMP primers are tested in reactions of different temperatures, a gradient ranging from 58 °C to 73 °C. The optimum results could be indicated by the highest turbidity reading or the brightest ladder-like bands in agarose gel electrophoresis. In many studies, positive results are usually recorded within the range of the optimum temperature (±5 °C), which depict the versatility of the LAMP primers and the *Bst* polymerase to amplify the target DNA without the need of an accurate thermal cycler [3,22]. This suggests that conventional water baths or heating blocks with minimal temperature control could be used for the LAMP method with reliable detection results, especially when running diagnostic tests in rural areas that lack access to sophisticated laboratory equipment.

The second parameter in optimizing this isothermal amplification is the reaction time. Many studies have reported the rapid output of LAMP results, in less than 30 min of reaction time in comparison with other established molecular detection methods such as PCR and qPCR [17,23,24]. In the expected result of ladder-like bands in agarose gel electrophoresis following the LAMP reaction, the intensity or brightness of the bands’ accumulation of the concatemers would increase over time due to the accumulation of the concatemers. The presence of excessive dNTPs in the reaction buffer, however, may trigger the chelation of magnesium ions [14,25]. Like any other nucleic acid amplification assay, the LAMP reaction would reach plateau stage after the exponential phase of rapid amplification. The concentration of dNTPs, template DNA or RNA, primers and polymerase units limits the nucleic acid amplification [14]. Hence, the amplification rate would decrease and eventually stop after any one of the limiters has been used up during the reaction [26].

#### 2.2.1. Analytical Sensitivity (Limit of Detection) of LAMP Method

The sensitivity or the minimum amount of target gene that may be detected and quantified is an important performance parameter for a molecular diagnostic system [27]. Typically, in any molecular screening including LAMP, an analytical sensitivity test is done to observe the detection limit of the assay with a minimum amount of target gene or positive control. The LAMP method has been proposed to have an excellent detection limit in analytical sensitivity, following comparison studies with other assays. Several researchers have identified the detection limit of the LAMP method at 0.01 to 10 plaque-forming units (pfus) of virus, which is 10 to 100 times more sensitive than the conventional PCR [2,28]. In a study on goose circovirus, the LAMP sensitivity was recorded to be at 100 pg of template DNA [24]. On the other hand, a finding in carryover contamination in DNA detection showed that the detection limit of the LAMP method was as low as 4 × 10^3^ copies of target DNA [29]. Research on LAMP sensitivity for authentication of several plant DNA molecules has shown a positive detection of around 10 fg, which is much more sensitive compared to other amplification methods [30].

#### 2.2.2. Analytical Specificity of LAMP Method

The presence of co-infected pathogens or a disease with like symptoms requires a screening system that is accurate enough to differentiate the target species while at the same time is not prone to false positive detection. MSA performed prior to the LAMP primer designing allows the recognition of the inclusive sequences within the target region in comparison with genes from related species. This follows the guidelines set by the US Food and Drug Administration (FDA) and Centers for Disease Control (CDC) on oligonucleotide primers used in diagnostics, in which the inclusivity of the target gene should be evaluated thoroughly by sequence analysis. To verify the applicability of the optimized LAMP method to specifically target the gene of interest, an analytical specificity test should be carried out against the closely-related species. The inclusion of several samples of the closely-related species as negative controls ensures that the LAMP method only detects the target gene with no cross reaction of false positive.

#### 2.2.3. Clinical Sensitivity (Percentage of True Positive) and Clinical Specificity (Percentage of True Negative)

The diagnostic performance of any molecular method can be analyzed via clinical sensitivity and specificity, which are also known as the percentage of true positive and the percentage of true negative, respectively, using clinical samples with defined results. The clinical sensitivity is measured by the percentage of the positive results obtained using the assessed method over the actual number of positive clinical samples. Vice versa, the clinical specificity is measured by the percentage of negative results recorded from the similar assessed method over the total of negative clinical samples. By adapting these clinical validations, the reliability of the LAMP method can be evaluated to receive approval by the FDA for on-field application and further development into diagnostic kits. The clinical sensitivity and specificity values signify the consistency of the LAMP method in comparison with the established screening protocols (qPCR, viral isolation) used to screen the clinical samples, where a value close to 100% suggests that the method can be validly utilized to replace the existing one. The formulae (Equations (1) and (2)) to calculate the clinical sensitivity and specificity using the LAMP method are as follows:

Formula (1) [31]:(1)$$\mathrm{LAMP}\;clinical\;sensitivity\;\left(Percentage\;of\;true\;positives\right)\;\;=\frac{The\;number\;of\;positive\;samples\;from\;LAMP\;method}{The\;number\;of\;positive\;clinical\;samples}\times 100\%\;$$

Formula (2) [31]:(2)$$\mathrm{LAMP}\;clinical\;specificity\;\left(Percentage\;of\;true\;negatives\right)\;\;=\frac{The\;number\;of\;negative\;samples\;from\;LAMP\;method}{The\;number\;of\;negative\;clinical\;samples}\times 100\%$$

### 2.3. Variation of LAMP Readouts for Field-Deployable Diagnostics

LAMP method is becoming the most studied and applied as compared to other isothermal amplification techniques, owing to the convenience of readily available reagents and guided by open-access publications in a wide range of biological fields. Even though the LAMP technology has been patented by Eiken Chemical Co., Ltd. (Tokyo, Japan) and developed into commercial kits, namely the Loopamp DNA Amplification Kit and Loopamp RNA Amplification Kit, several life-science companies have invested in the licensing to develop their own LAMP reagents, such as the WarmStart^®^ LAMP kit from NEB (Hitchin, UK), GspSSD2.0 Isothermal Mastermix from OptiGene (Horsham, UK) and LavaLAMP™ DNA Master Mix from Lucigen (Middleton, WI, USA). Such LAMP reagents can be easily handled by non-expert personnel without the reliance on specific instruments, which promotes more novel discoveries by scientists in diagnostic studies. Previously, LAMP results were analyzed by the presence of ladder-like bands following agarose gel electrophoresis, which is not convenient for on-site detection. In many applied innovations, LAMP nowadays is demonstrated to be compatible with a variety of signal readouts such as colorimetric, fluorescence, magnetic bead aggregation and turbidimetric [32,33].

Turbidity or cloudiness of the post-LAMP reaction is based on the precipitation of magnesium pyrophosphate as a by-product of the LAMP reaction [34]. The accumulation of precipitates can be visually observed by naked eyes at the bottom of the reaction tube, serving as a simple and direct readout method. The level of the turbidity can be relatively quantified in a real-time detection using a commercial turbidimeter. Such turbidity-based readouts, however, are reported to be less sensitive in detecting low amounts of template as compared to fluorescence detection, which has displayed 10 times greater sensitivity [35].

Intercalating DNA dyes such as PicoGreen, SYBR Green and SYTO fluoresce with UV illumination for clear and sensitive LAMP detection on a minuscule amount of template [36,37]. The LAMP reaction that proceeds can be monitored in real time using a fluorescence detector. Nevertheless, the protocol with post-LAMP addition of the fluorescence dye may introduce aerosol contamination and leads to the occurrence of false positives from opening the reaction tube [18]. One alternative to overcome such issue is the application of colorimetric metal-sensitive indicators such as single-step detection by direct visualization of color changes. Colorimetric LAMP protocols have been developed by adding either one of these indicators and observing the shift in solution color (negative to positive) with naked eyes; hydroxy naphthol blue (violet blue to sky blue), calcein (dark yellow to yellow) and malachite green (colorless to light blue/greenish blue), to mitigate the risk of contamination and shorten the overall process [37,38,39,40]. The changing in color of hydroxy naphthol blue and calcein sometimes may be difficult to distinguish prior to optimization of the reaction cofactors, while moderate sensitivity of a minimum 100 target copies has been reported [41].

In any nucleic acid amplification, the formation of new phosphodiester bond triggers the release of protons from the sugar-phosphate backbone of DNA and hence lowers the solution pH to a slightly acidic condition [42]. A typical polymerase in PCR reaction requires a constant and optimal pH level, aided with buffer solution. In the LAMP method, the polymerase with strand displacement ability has shown good tolerance in a weakly to zero buffered reaction mixture in a pH range of 6.0 to 10.0 [41]. Combining the tolerance of the polymerase and pH change throughout amplification, such properties can be exploited for the application of a pH-sensitive detector or sensor. The changes in pH of the reaction mixture as the isothermal amplification proceeds can be observed by adding pH-sensitive dyes for direct visual detection such as bromothymol blue, bromocresol purple, cresol red and phenol red [41,43]. These broad spectrums of colorimetric indicators facilitate rapid LAMP detection of target pathogens at low production cost without impeding the sensitivity and specificity of the assay. Overall, optimized workflow with clinical tests validation would ensure the developed LAMP method is reliable for field application in point-of-care testing of various pathogens (Figure 2).

## 3. LAMP Application in Screening of Avian Viruses

### 3.1. Detection of Avian Influenza Viruses of Multiple Subtypes

Avian influenza viruses (Influenza A virus) are the highly infectious viruses that have affected the poultry industries for many years. The highly pathogenic avian influenza (HPAI) virus has caused severe economic damage to the poultry industry with up to 100% mortality. For example, the outbreak of avian influenza H7N1 in Italy caused more than 13 million chicken deaths from 1999 to 2000 [44]. The affected hosts are not only food producing birds such as the chicken, quail and turkey, but also pet birds and wild birds. As a result of high vulnerability and mortality in chickens due to the HPAIs, monitoring and screening against such viruses are crucial to secure the essential poultry source and control the spread of the avian influenza diseases towards susceptible environments. Sporadic cases of zoonotic infection of animal-to-human transmission that ended up with mortality have also been reportedly caused by avian influenza virus subtypes H1N1, H2N2, H5N1, H7N7 and H7N9 [5,6].

Influenza A virus is the only species that belongs to the genus of *Alphainfluenzavirus*, with major natural reservoirs in birds and some mammals [45]. Many subtypes of the Influenza A virus have been discovered, and classifications are made according to combinations of the different viral glycoproteins hemagglutinin (H) and neuraminidase (N). Among the highly pathogenic subtypes are H5 and H7 [44]. These RNA viruses have a total of around 13.5 kb of genome that are divided into eight RNA segments. The efforts in screening of influenza A viruses have been crucial to control the spread of the pathogens while mitigating the losses caused. Together with conventional and sensitive screening methods such as qPCR, RT-qPCR, nucleic acid sequence-based amplification (NASBA) and next-generation sequencing (NGS), the LAMP method has been applied for genetic detection of the highly pathogenic avian viruses [45].

The widespread avian influenza virus subtype H5N1 has been accurately detected using the LAMP method in many studies [5,44,46,47,48,49,50,51]. The detection of avian influenza virus subtype H7 using the LAMP method has shown a promising sensitivity at a level of 0.01 PFU/tube in the assay, which was 100-fold more sensitive than RT-PCR in a comparison study [44]. The sensitivity obtained in the H7-RT-LAMP method is in agreement with the research on LAMP diagnosis against avian influenza virus subtype H5 [47]. The detection limit of the LAMP method in detecting the LPAI H7 subtypes was also as good as that of HPAI [44]. Clinical samples of throat and cloacal swabs have higher predictive value in surveillance screening of highly pathogenic influenza viruses [44,46]. From a cultured specimen containing avian influenza virus subtype H9, the detection rate of confirmed positive samples using RT-LAMP was recorded at 100%, while RT-PCR had a lower positive rate at 91.8% due to several false negative results [46]. The detection limit of the H9 subtype was recorded at 10 copy numbers, which is tenfold more sensitive than that of RT-PCR [46]. The RT-LAMP method developed for the detection of avian influenza subtype H7N9 showed that the assay is highly specific with sensitivity of up to 50 copies/reaction and applicable for a direct RT-LAMP without the additional nucleic acid extraction step [5]. The high sensitivity level of the RT-LAMP method in the previous avian influenza studies indicates that this method is viable for the detection of the early stage or newly acquired infections with low viral amounts. Recently, a set of LAMP primers designed on the matrix (M) gene of *Alphainfluenzavirus* was able to screen all subtypes of avian influenza viruses, which included positive detection of subtypes H1, H5 and H9 from clinical specimens [50]. The broad-range LAMP method has recorded a faster detection time in all reactions as compared to the previous LAMP method targeting the matrix (M) gene [52].

### 3.2. Screening of Polyomaviruses in Birds

Polyomaviruses of birds under the genus *Gammapolyomavirus* or *Avipolyomavirus* consist of several species of virus that are host-specific to different species of birds. Among the recorded species of avian polyomaviruses in the virus taxonomy are budgerigar fledgling disease virus, Adelie penguin polyomavirus, butcherbird polyomavirus, canary polyomavirus, crow polyomavirus, finch polyomavirus and goose hemorrhagic polyomavirus [53,54,55,56]. Known to cause diseases and mortality in many species of birds, this group of avian polyomaviruses carry a circular genome of around 5 kb of double-stranded DNA in a non-encapsulated icosahedral capsid [54,57]. Pathological symptoms of birds infected with avian polyomaviruses, following clinical analysis and histopathologic lesions, include hepatitis, hydropericardium and ascites [53,58]. Fledgling or young birds are the main groups affected by avian polyomaviruses with high morbidity and mortality rates, and the positive cases are more prominent with birds kept in captivity with close proximity [58,59].

The incidences of goose hemorrhagic polyomavirus (GHPyV) infection have caused economic losses in waterfowl production at large scale, with a 32% mortality rate of young goslings reported in Poland [56,60]. Adult parrots infected with budgerigar fledgling disease virus (Aves polyomavirus 1, APyV) are more resistant to the inflammatory diseases and while they appear to be asymptomatic, the virus can be shed via feathers and faeces for up to 6 months [61,62]. The fact that the virus can survive in an outside environment for many months raises concerns for swift yet efficient measures such as early screening, cage sanitization and isolation of the infected birds. Budgerigar fledgling disease virus is also observed to co-infect parrots together with beak and feather disease virus (BFDV) in many cases [63,64,65]. Hence a diagnostic method that is able to segregate the existence of different viruses is necessary to give specific treatments for the infected birds.

Among the developed diagnosis methods for the molecular detection of different avian polyomaviruses are LAMP, PCR, qPCR and enzyme-linked immunosorbent assay (ELISA) [56,59]. A study on the detection of goose hemorrhagic polyomavirus in geese and ducks showed the LAMP method has a good sensitivity in limit of detection test at 1.5 pg of extracted DNA, despite qPCR displaying 100-fold more sensitive results [56]. Positive detections were recorded from clinical specimens with diseased symptoms, indicating the consistency of the LAMP method in detecting the virus as confirmed by the qPCR results using the similar specimens [56]. To verify the possibility of any cross-reactivity, the optimized LAMP method was tested against GHPyV with other pathogens present in ducks and geese to observe the potential of cross-reactivity that might lead to a false positive result. Prevalent infections of budgerigar fledgling disease virus (APyV) among parrots are concerning, and should not be neglected in the pet bird industry as well as the global parrot population, which is why the LAMP method has been recently developed for efficient screening of this virus [61]. Using hydroxyl naphthol blue (HNB) in the reaction mixture for direct monitoring of the results, the colorimetric LAMP method on budgerigar fledgling disease virus has shown 100% sensitivity and specificity in diagnostic performance [61]. The exclusivity of the developed LAMP method was also assessed using different avian pathogens, including the BFDV that commonly co-infects parrots [61]. The limit of APyV detection was observed at 500 copies/reaction for both LAMP and PCR methods [61]. The analytical sensitivity results of LAMP being as great as, if not greater than, those from qPCR suggests that the LAMP method is satisfactory to replace the standard diagnostic as a handy and cheaper alternative, especially for diagnostic application in small veterinary clinics or on-field screening.

### 3.3. Diagnosis of Circoviruses in Different Avian Hosts

Circoviruses are among the cluster of small viruses in the Circoviridae family, with non-enveloped icosahedral virions of approximately 20 nm in diameter. Enclosed within the virions are circular genomes of single-stranded DNA, with sizes ranging from 1.8 kb to 2.2 kb according to the species that infect different mammalian hosts. Within the avian hosts, infections of circoviruses have been reported in the goose, canary, pigeon, swan, duck and parrot [23,24,66,67,68]. Circoviruses in avian hosts are identified to cause immunosuppression, making the hosts much more vulnerable to secondary infections and eventually leading to mortality [23,67,68].

Goose circovirus (GoCV) is often associated with runting and stunting syndromes in the commercial geese flock with a high degree of mortality [24,67]. Feather disorders are the non-specific clinical symptoms while pathological studies revealed hemorrhages and splenomegaly [67]. The virus also causes T-lymphocyte depletion in lymphoid organs, which results in an immunosuppressive effect [67]. Screening of this virus via the LAMP method offers early detection to swiftly isolate the infected birds from the remaining flock, especially when asymptomatic infections occur [24]. The simple diagnosis method was developed by specifically targeting the Vgp4 gene of the GoCV, which codes for one of the viral capsid proteins. The LAMP method on goose circovirus showed a detection limit at 100 pg DNA equal to the qPCR method, while no false positive was observed with DNA samples of other avian viruses infecting geese such as goose hemorrhagic polyomavirus or goose parvovirus [24]. Field samples with clinical symptoms were tested positive at 97.4% sensitivity using the optimized LAMP method in comparison study with qPCR assay [24].

As the name itself suggests, beak and feather disease virus (BFDV) has been identified to cause severe feather loss and abnormal beak shape as the significant symptoms of this circovirus infection in parrots, similar to the goose circovirus symptoms [23,69]. BFDV infection is the most common viral disease affecting parrots, highly transmittable horizontally and vertically to more than 60 different species of psittacine [70]. A LAMP study has demonstrated an excellent limit of detection against BFDV at 3.5 fg of viral DNA following optimal reaction temperature at 63 °C [23]. Clinical specimens of tissues and livers from parrots suspected to be infected with BFDV were tested with both conventional PCR and optimized LAMP methods, where the LAMP method showed greater sensitivity by detecting more positive samples [23]. The application of this simple sample pre-treatment method highlights the feasibility of LAMP in detecting target pathogens in different types of samples for a rapid diagnostic procedure. In a more recent development, the LAMP reaction has been incorporated with swarm primers to reduce the reaction time of BFDV detection to 40 min of complete reaction time and the results can be directly visualized with color changes of the hydroxy naphthol blue dye [71]. LAMP with swarm primers also demonstrated a great improvement in diagnostic performance by showing 100% clinical sensitivity from 71% of the standard LAMP method, following BFDV detection from psittacine samples of blood, feather, tissue and cloacal swabs [71]. Early and rapid diagnosis using the LAMP method to screen infected parrots prior to any international trade would help to control virus transmission [71].

### 3.4. Screening of Immunosuppressive Viruses in Chickens

The economically endangering chicken anemia virus (CAV), or Gyrovirus, was once grouped within the family of Circoviridae until a recent taxonomical classification was made to classify the virus into the family of Anelloviridae due to significant differences in genome organization [72,73]. As a sole species under the Gyrovirus genus, this single-stranded DNA virus has caused the infectious chicken anemia disease that affects the poultry industry worldwide [74]. Similar to the viruses from the Circoviridae family, CAV is known to cause immunosuppression of the host by replicating inside the cortex thymus to destroy the precursor T cells [73]. The immunosuppressed chickens are more likely to acquire secondary infections that increase morbidity and mortality, which ultimately cause economic loss. Another replication site of the CAV is in hemocytoblast of the bone marrow, where the destruction of these stem cells leads to aplastic anemia [73]. Epidemiological studies showed that CAV infects a majority of chicks as young as 1 day old, making them the most vulnerable group via the vertical transmission of hatching eggs [75]. Meanwhile horizontal transmissions can occur via the fecal–oral route and diseased feather follicle epithelium from the infected litter within flocks [73]. Complete eradication of CAV in contaminated cages is less likely due to the virus having high resistance to heat and chemical disinfectants [73]. While currently there is no proper medication to specifically treat CAV infection, a common approach in controlling the viral outbreak is through vaccination of the breeding chickens prior to the stage of egg production [73,75]. Immunological assays, conventional PCR and viral isolation are the diagnostic methods used to screen for the presence of CAV [75,76]. Though viral isolation is not often applied for diagnosis due to the meticulous procedures needed together with high cost, a novel LAMP detection of CAV was proposed as a simple and inexpensive alternative. The LAMP method targeting the VP2 gene of CAV was able to detect template DNA at a minimum amount of 100 fg within 30 min of reaction [75]. CAV DNA from the liver of an infected broiler chicken was successfully distinguished by the LAMP method, corresponding to the clinical result from the conventional PCR method [75]. This method can be adapted as a suitable tool for occasional surveillance and epidemiological studies of CAV infection.

Marek’s disease virus (MDV) or the scientific name Gallid herpesvirus 2, a pathogenic and oncogenic serotype 1 of the family Herpesviridae, is one of the most infectious pathogens in poultry production [77]. It was reported that ubiquitous MDV infections occurred frequently in more than 70 countries, including the emergence of very virulent strains (vvMDV+) [78]. Upon entry via inhalation of infected dusts, MDV is transmitted to B cells and T cells for viral replication to take place, which may cause cell death [79]. The infected cells reach latency phase to proliferate abnormally and spread within the host, causing tumors in the liver, spleen and kidney [77,79]. The vaccination programs for MDV have reported an increase in demand for the protection against this avian virus, including double dosage and revaccination, which highlights the urgency to tackle Marek’s disease globally [78]. Several studies have utilized the LAMP method in detecting the presence of MDV, where the findings highlight efficiency and accuracy of this simple method as the major advantages in handling viral diagnosis [76,77,80,81]. Clinical evaluation has shown a positive detection rate at 95%, with the limit of detection as low as 50 copies [76,80]. In a very practical method, the dust of fine skin particles and feather debris from chickens infected with MDV were able to produce positive detection using a simple heat treatment method [77].

Another immunosuppressive disease affecting the poultry industry is infectious bursal disease (IBD), instigating morbidity of almost 100% in susceptible flocks while very virulent strains are deadly enough at a mortality rate of around 60% [76,82]. IBD is also known as Gumboro disease and infectious bursitis, caused by the only species of the Avibirnavirus genus, namely infectious bursal disease virus (IBDV) [83]. This double-stranded RNA virus is most lethal towards young chicks of age between 3 to 6 weeks old at lower antibody levels [8,76]. Pathogenic Serotype 1 of IBDV destroys the B lymphocytes in the chicken’s bursa, impeding the host humoral immunity and consequently becoming more vulnerable to secondary infections by other pathogens [8]. RT-LAMP tested against the IBDV positive clinical samples demonstrated 100% sensitivity and specificity [8,76]. Using a biotin-labeled DNA probe designed to target the amplified RT-LAMP products, a lateral flow dipstick was applied for the molecular detection of IBDV [8]. FITC-labeled probes were added following the RT-LAMP reaction for the hybridization to occur with the DNA probes. The positive result could be visualized using naked eyes by either the turbidity of the white precipitate or the dark purple band developed on the test line of the dipstick [8].

### 3.5. Wide Application of LAMP against Notable Viruses in Poultry Industry

Apart from these highlighted avian viruses that have been successfully screened using the sensitive yet rapid LAMP detection, this method has been widely utilized in many other viruses that adversely affect the poultry industry. Such LAMP application includes distinguishing the subgroup of avian leukosis virus (ALV) from other exogenous Alpharetrovirus subgroups [84,85]. The ALV subgroups are segregated based on the glycoproteins of the viral envelope, a major factor that regulates antigenicity, host range and the occurrence of viral interference among the group [85]. While all the ALV field strains are oncogenic, the neoplastic lymphoid leukosis disease triggered by this cluster of RNA viruses on chicken flocks needs to be identified and addressed specifically according to the subgroup. Infectious bronchitis virus (IBV) is another major cause of respiratory disease in chicken farms that has been underlined with the urgency of rapid diagnosis via the optimized LAMP method [16,86,87]. With single-stranded RNA avian coronavirus as the causative agent, from the Coronaviridae family, IBV was first tested with RT-LAMP at the conserved regions of the nucleocapsid gene [16]. Through an innovative mRT-LAMP-LFD, a multiple LAMP method was coupled with a lateral flow dipstick for simultaneous detection of IBV and the contagious Newcastle disease virus (NDV) [87]. In the diagnostic tests, the presence of IBV and NDV on clinical samples detected using the mRT-LAMP-LFD assay were at decent sensitivity rates of 98.65% and 97.25%, respectively. Single-step RT-LAMP on IBV that relies on fluorescence emissions was developed to facilitate semi-quantification of the virus spread at molecular level in veterinary laboratories which lack mainstream instruments [86].

Besides chickens, ducks are also heavily affected by avian viruses in large-scale poultry production. Duck viral hepatitis (DVH) is a lethal and highly contagious disease typically affecting young ducklings. Known to be associated with substantial liver lesions with a concerning death rate, the most widespread viral strain is named duck hepatitis A virus type-1 (DHAV-1) from the Picornaviridae family. The presence of this RNA virus was distinguished through an established RT-LAMP method by targeting the conserved regions in the 3D gene, with a detection limit as low as 0.3 pg [14,88]. The RT-LAMP method was also used to detect Tembusu virus (TMUV), a species of Flavivirus virus that causes large economic loses in egg-laying and breeder duck farms, especially in China [89,90]. This virus species from the Flaviviridae family is associated with many forms of transmissions, such as mosquito borne, airborne, direct contact and vertical transmission, which could all be regulated through point-of-care detection at infected farms using the viable RT-LAMP method [89,91,92]. Advocated as a minimal technology analytical tool, routine screening in poultry farms would surely be of benefit to regulate any virus transmission to avian hosts.

As an overview of the LAMP application in screening prominent viruses in the poultry industry and domestic birds, the examples of the detection limit as well as clinical sensitivity for each of the LAMP methods developed against the respective viruses are listed in Table 1.

Figure 3 summarizes the viruses (in short form) detected using the LAMP method as discussed throughout the review under different groups of the avian hosts; chicken, duck, goose and parrot.

## 4. Summary and Conclusions

Throughout the decades, disease epidemics persist as the main problem in the poultry sector internationally. Controlling the spread of avian viruses in the poultry industry is a great and worldwide responsibility as it is a major food source. Similar necessities are pertinent to the international trade of pet birds, as well as sustainability of local breeding and wild birds. This is in line with the guidelines set by the World Organization for Animal Health, which emphasize the importance of biosecurity measures in safeguarding poultry production and trade while protecting food security and farmers’ livelihoods. While the efforts in fighting these pathogenic biological agents should be continuous, it is also crucial to stay alert to the potential reintroduction of viruses into different susceptible avian species, or even across the kingdom border such as the infection of humans. Since chickens are known to be a reservoir for various infectious pathogens, the consumption of contaminated meat and eggs could possibly transmit such pathogens to humans.

The conventional virus isolation method is known to be accurate and, in some cases, more sensitive than the LAMP method in screening the presence of avian virus. This technique is, however, more expensive and time consuming, taking several days to more than a week to get a result. Despite the fact that sophisticated qPCR can be performed much faster (2-3 h), the equipment required to run the assay is costly and less accessible in isolated areas. In the course of diagnosis, it is important to identify the cause of infection as soon as possible so that control plans can be administered swiftly to avoid further spread in the infected group. For a long-term solution, vaccines are now available for prevention against several avian viruses such as avian influenza virus, chicken anemia virus and Marek disease virus. The downsides of vaccinations on the other hand should be taken into consideration, especially in large-scale farming of poultry. These issues, including a surge in production costs, poor vaccine administration, emergence of new virus variants and unproductive effects of vaccines, all affect the poultry industry today.

Many benefits of the LAMP method have been well justified yet the cons that often arise from this isothermal assay should be recognized and resolved properly. Since the LAMP method is very sensitive in detecting very low amounts of template, the occurrences of false positive results have been frequently observed in many diagnostic studies. Among the factors that contribute to the false positive results is the low incubation temperature, causing non-specific binding of the primers to the target sequence, while *Bst* polymerase can readily amplify the hybridized structure at a temperature as low as 55 °C. Excessive free dNTPs and Mg2+ ions are associated with non-target amplification, as reported in previous studies [93]. While the addition of betaine mitigates the formation of secondary structures, too high a concentration of betaine is shown to reduce the amplification efficiency [94]. The optimization of the LAMP reaction as well as the reagents used must be prioritized accurately to avoid such problems.

It is also important to note that the use of multiple primers within a reaction may interfere with the aligning process due to an increase in competition between one and another. The standard ratio for inner primers (FIP and BIP) to outer primers (F3 and B3) is 4:1. With the inclusion of loop primers, the concentration of inner primers, loop primers and outer primers could be set at a ratio of 8:2:1 respectively. An incompatible ratio between the LAMP primers will affect the amplification sensitivity, hence finding the right amount is important during the early assay development.

In the protocol of RT-LAMP that requires the incubation for reverse transcriptase activity as an additional step, the whole process would consume more time and cost. The utilization of extra reagents and sample handling steps might induce a greater chance of contamination and lead to false positive detection. Besides, most of the LAMP results are reported as qualitative data for either positive or negative detection based on color changes and gel electrophoresis. Only relative quantification of the turbidity and spectrophotometric evaluation from LAMP products can be measured according to the subjected samples. Therefore, the LAMP method is more suitable for early screening in diagnostics, preceding to in-depth research on the target pathogen.

As emphasized in this review, simple nucleic acid extraction techniques combined with a fast molecular assay would serve as a convenient screening strategy in combating pathogenic avian viruses. From the examples of established applications in different avian viruses, the LAMP method can be adapted as a fast point-of-care screening for the designated viruses. A more advanced approach would be to miniaturize LAMP reactions and integrate them on a chip for a more convenient approach. Microfluidic and digital LAMP-on-a-chip systems have been demonstrated to perform single gene detection, multiplex-gene detection and RT-LAMP [95]. However, careful consideration should be made of the cost of sensor or biosensor development for mass production against avian diseases.

At the current progress, LAMP could be readily applied at any under-equipped veterinary hospitals and private veterinary clinics. Obtaining a fast result from the screening is proportional to the swift and efficient control of the pathogen spread. Infected hosts could be isolated or quarantined as soon as possible, and treatments administered specifically against the screened pathogens. While the occurrence of contaminations or false positives is common with LAMP due to the surplus in sensitivity, innovations in single-step techniques and direct visualizations might be the answer needed to improve the method towards on-farm application. Judging from the trend of advancements in molecular diagnostics that are widely integrated with modern technologies, point-of-care testing at farms with the LAMP method is highly likely to be developed and accessible by even the farmers themselves in the near future.

## Figures and Tables

**Figure 1 animals-12-00076-f001:**
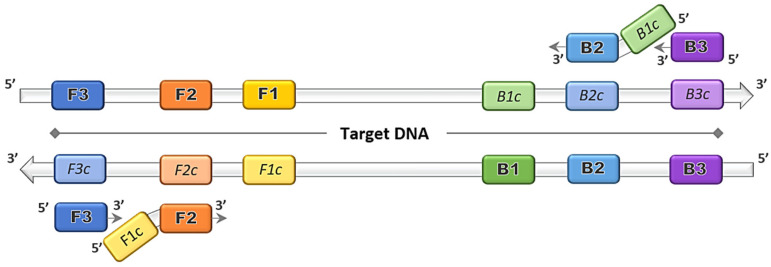
Loop-mediated isothermal amplification (LAMP) primers binding on target DNA. Four principal LAMP primers namely F3, FIP (F2 + F1c), B3 and BIP (B2 + B1c) bind on the complementary target regions and amplifications proceed in the direction of 5′ to 3′.

**Figure 2 animals-12-00076-f002:**
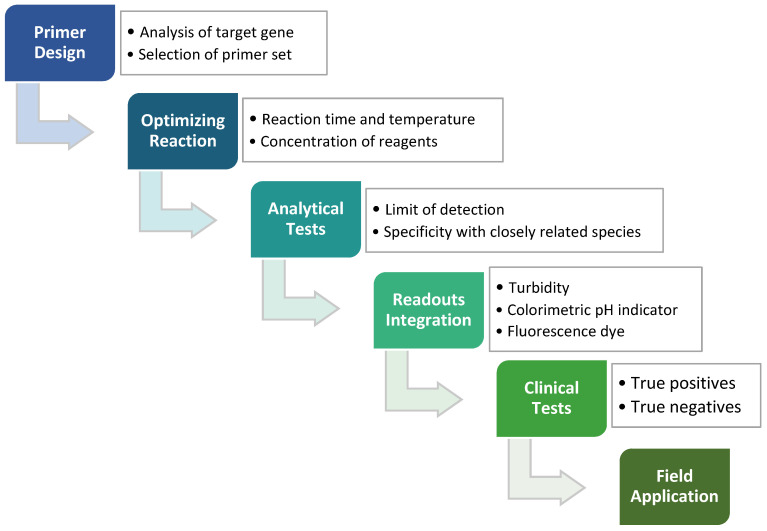
The process of developing the LAMP method, which starts from the designing of LAMP primers against target genes, followed by the reaction optimizations, analytical tests, readouts integration and lastly clinical tests. An applicable LAMP method can eventually be applied to field diagnostics such as screening routines or the production of LAMP kits.

**Figure 3 animals-12-00076-f003:**
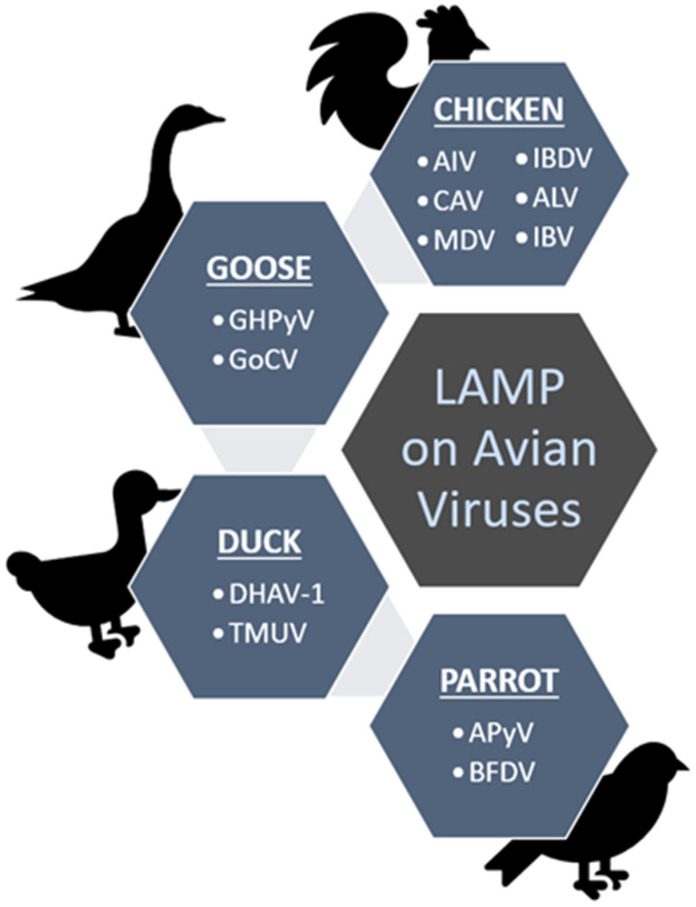
Summary of the discussed avian viruses that have been screened from chickens, geese, ducks and parrots using the established LAMP methods.

**Table 1 animals-12-00076-t001:** LAMP applications in the detection of avian viruses from different avian hosts with optimal limit of detection and clinical sensitivity in diagnostic performance.

Avian Host	Virus	LAMP Limit of Detection	LAMP Clinical Sensitivity
Chicken	Avian influenza virus (AIV)	0.01 PFU/µL [44] 0.1 PFU/µL [48,50] 10 copies [46]	100% [46,50]
	Chicken anemia virus (CAV)	100 fg [75] 50 copies [76]	100% [76]
	Marek’s disease virus (MDV)	20 copies [76]	100% [76] 95.0% [80]
	Infectious bursal disease virus (IBDV)	250 copies [76]	100% [8,76]
	Avian leukosis virus (ALV)	5 copies [85] 20 copies [84]	98.0% [85]
	Infectious bronchitis virus (IBV)	1 EID50/mL [86] 6.3 copies [87]	100% [86] 99.5% [16] 98.7% [87]
	Newcastle disease virus (NDV)	5 copies [87]	97.3% [87]
Duck	Duck hepatitis A virus (DHAV-1)	0.3 pg [14,88]	100% [88]
	Tembusu virus (TMUV)	100 fg [89] 20 copies [89]	97.5% [89]
Goose	Goose hemorrhagic polyomavirus (GHPyV)	1.5 pg [56]	100% [56]
	Goose circovirus (GoCV)	3.5 fg [24]	97.4% [24]
Parrot	Budgerigar fledgling disease virus (APyV)	500 copies [61]	100% [61]
	Beak and feather disease virus (BFDV)	3.5 fg [23] 100 copies [71]	100% [71]

## Data Availability

Not applicable.

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
