# Peer review of "Loop-Mediated Isothermal Amplification (LAMP) as a Promising Point-of-Care Diagnostic Strategy in Avian Virus Research"

_animals, 2021, doi:10.3390/ani12010076_

Round 1
Reviewer 1 Report
The following comments may help the authors to improve the manuscript before an acceptance.
- The authors should provide references for formula 1 and 2 in pages 6 (section 2.2.3)
- For field-deployable diagnostics, the concept of point-of-care (PoC) devices should be mentioned to help the readers having a broader context. See: https://doi.org/10.1016/j.trac.2020.116004
- Would lab-on-a-chip technology play a role in the on-site detection using LAMP technology?
Author Response
Point 1: The authors should provide references for formula 1 and 2 in pages 6 (section 2.2.3). 

Response: Thank you for the suggestion. Te references for formulae 1 and 2 have been added in the text (section 2.2.3, lines 252 and 255), and reference list has been updated.
Point 2: For field-deployable diagnostics, the concept of point-of-care (PoC) devices should be mentioned to help the readers having a broader context. See: https://doi.org/10.1016/j.trac.2020.116004
Response: Thank you for the comment. More description of point-of-care testing has been mentioned in lines 83-94 to assist the readers.
…... “The rapid spread of highly pathogenic avian viruses has brought major concerns on the need of a quick detection system, in which the LAMP method is proposed as a convenient solution via point-of-care testing, i.e. an analysis that can be performed on-site or close to the desired setting. Point-of-care testing was previously dominated by biosensor and lateral flow strips or dipstick using immobilized labels and antibodies. Nonetheless, emerging utilization of polymerase chain reaction (PCR) has led to the development of several isothermal amplification technologies such as LAMP, recombinant polymerase amplification (RPA), strand displacement activity (SDA), nucleic acid sequences-based amplification (NASBA), helicase-dependent amplification (HDA) and rolling circle amplification (RCA). Nucleic acid-based systems are considered simpler, cost-efficient, fast, accurate, reliable and easy to operate even by a non-technical personnel. These are anticipated merits of an ideal point-of-care diagnostic strategy. Throughout the establishment of these methods, many common avian viruses have been successfully screened via clinical studies with good limit of detection and efficient sensitivity reported.”
Point 3: Would lab-on-a-chip technology play a role in the on-site detection using LAMP technology?
Response: Thank you for the question triggered. Yes, a LAMP system can be miniaturised and integrated as lab-on-a-chip for a more convenient on-site detection. We added brief elaboration regarding this in lines 628-633.
….. “A more advanced approach would be to miniaturise LAMP reactions and integrate them on a chip for a more convenient approach. Microfluidic and digital LAMP-on-a-chip systems have been demonstrated to perform single gene detection, multiplex-gene detection and RT-LAMP [94]. However, careful consideration should be made on the cost of the sensor or biosensor development for mass production against avian diseases.
Reviewer 2 Report
It is recommended to add some content, such as the disadvantages of LAMP technology: interference between primers, high false positive rate
Author Response
Point 1: It is recommended to add some content, such as the disadvantages of LAMP technology: interference between primers, high false positive rate.
Response: Thank you for the suggestion. The anticipated content has been added in lines 596-623. Reference list has been updated.
…… “Many benefits of the LAMP method have been well justified yet the cons that of-ten arise from this isothermal assay should be recognized and resolved properly. Since the LAMP method is very sensitive in detecting very low amount of template, the oc-currences of false positive results have been frequently observed in many diagnostic studies. Among the factors that contribute to the false positive results is the low incu-bation temperatures, causing non-specific binding of the primers to the target se-quence while Bst polymerase can readily amplify the hybridized structure at the tem-perature as low as 55 °C. The excessive free dNTPs and Mg2+ ions are associated to non-target amplification, as reported in previous studies [92]. While the addition of betaine mitigates the formation of secondary structures, too high concentration of be-taine is shown to reduce the amplification efficiency [93]. The optimisation of the LAMP reaction as well as the reagents used must be prioritised accurately to avoid such problem.
It is also important to note that the use of multiple primers within a reaction may interfering the aligning process due to increase in competition between one and an-other. The standard ratio for inner primers (FIP and BIP) to outer primers (F3 and B3) is 4:1. With the inclusion of loop primers, the concentration of inner primers, loop pri-mers and outer primers could be set at a ratio 8:2:1 respectively. Incompatible ratio between the LAMP primers will affect the amplification sensitivity, hence finding the right amount is important during the early assay development.
In the protocol of RT-LAMP that requires the incubation for reverse transcriptase activity as additional step, the whole process would consume more time and cost. The utilisation of extra reagents and sample handling steps might induce greater chance of contamination and lead to false positive detection. Besides, most of the LAMP results are reported as qualitative data for either positive or negative detection based on color changes and gel electrophoresis. Only relative quantification of the turbidity and spec-trophotometric evaluation from LAMP products can be measured according to the subjected samples. Therefore, the LAMP method is more suitable for early screening in diagnostics, preceding to in-depth research on the target pathogen.
Round 2
Reviewer 1 Report
- a definition of point of care device/testing should be provided (comment number 2 in previous review).
Author Response
Thank you for the comment. More description of point-of-care testing has been mentioned in lines 83-94. The suggested paper https://doi.org/10.1016/j.trac.2020.116004 has been cited as [7] to further guide the readers to additional reading.
…... The rapid spread of highly pathogenic avian viruses has brought major concerns on the need of a quick detection system, in which the LAMP method is proposed as a convenient solution via point-of-care testing, i.e. an analysis that is conducted on-site or close to the desired setting or patient. Point-of-care testing was previously dominated by biosensor and lateral flow strips or dipstick using immobilized labels and antibodies. Nonetheless, emerging utilization of polymerase chain reaction (PCR) has led to the development of several isothermal amplification technologies such as LAMP, recombinant polymerase amplification (RPA), strand displacement activity (SDA), nucleic acid sequences-based amplification (NASBA), helicase-dependent amplification (HDA) and rolling circle amplification (RCA). Nucleic acid-based systems are considered simpler, cost-efficient, fast, accurate, reliable and easy to operate even by a non-technical personnel [7]. These are anticipated merits of an ideal point-of-care diagnostic strategy. Throughout the establishment of these methods, many common avian viruses have been successfully screened via clinical studies with good limit of detection and efficient sensitivity reported.